# Floristic Legacies of Historical Land Use in Swedish Boreo-Nemoral Forests: A Review of Evidence and a Case Study on *Chimaphila umbellata* and *Moneses uniflora*

**Ove Eriksson**

Department of Ecology, Environment and Plant Sciences, Stockholm University, 10691 Stockholm, Sweden; ove.eriksson@su.se

**Abstract:** Many forests throughout the world contain legacies of former human impacts and management. This study reviews evidence of floristic legacies in the understory of Swedish boreo-nemoral forests, and presents a case study on two currently declining forest plants, suggested to have been favored by historical use of forests. The review provides evidence of forest remnant populations of 34 grassland species. Thus, many floristic legacies have their main occurrence in semi-natural grasslands, but maintain remnant populations in forests, in some cases more than 100 years after grazing and mowing management have ceased. Despite less information on true forest understory plants appearing as legacies of historical human use of boreo-nemoral forests, a putative guild of such species is suggested. The case study on two species, *Chimaphila umbellata* and *Moneses uniflora* (Pyroleae, Ericaceae) suggests that both species are currently declining, mainly due to modern forestry and ceased livestock grazing in forests. *Chimaphila* maintains remnant populations during decades, due to its extensive clonal capacity and its long-lived ramets. *Moneses* is more sensitive, due to a lower stature, weaker clonal capacity and short-lived ramets, flowering only once during their lifetime. Thus, *Moneses* have more transient occurrences, and will decline rapidly under deteriorating conditions.

**Keywords:** historical land use; forest biodiversity; livestock grazing; partial mycoheterotrophs; remnant populations





## 1. Introduction

Historical legacies of former human activities are common in forests throughout the world, e.g., [1–4]. In addition to material remains of these activities, associated with for example settlements, e.g., [5–8], these legacies may be manifested biologically. Many species, especially plants, remain as a historical signal reflecting past human activities and management regimes, e.g., [9–12]. This holds true also for boreal forests in Scandinavia, particularly for the southern transition zone, usually termed the boreo-nemoral zone [13], which harbored the largest human population historically. Historical legacies in present-day boreal and boreo-nemoral tree communities are well-known and have been recognized in several studies, e.g., [14–16]. In contrast, legacies in the boreal and boreo-nemoral forest understory have been focus of less research [17], for example in comparison with still remaining open habitats such as pastures and meadows, e.g., [18]. Generally, ecological perspectives on historical use of forests are often overlooked, not only in Scandinavia but all over Europe [19].

Boreo-nemoral forests in Scandinavia have been much influenced by humans ever since the arrival of agriculture around 4000 BC [20]. The impact has since that time more or less continually increased [21], although interrupted by periods of population decline, for example due to wars, famine and disease [22]. Forests were cleared and burned to make room for agriculture, i.e., crop fields and hay-meadows for production of winter fodder, and forests were used for livestock grazing. Slash-and-burn cultivation was practiced

regionally [23,24]. Controlled burning was also used to improve grazing conditions in forests [25]. Production of winter fodder was conducted not only in open meadows but also by harvesting twigs and leaves, both in remaining forested areas [26] and in wooded meadows [27,28]. Wetlands, i.e., mires, bogs and shores along streams and lakes were particularly important for production of winter fodder [29,30]. Livestock grazed in forests outside the infields of the farms [31–33]. The impact of livestock was massive [34]. The forests in southern Sweden during the 19th century have been described as "a large cow pasture" [35] (p. 395). In addition, forests were used for production of charcoal, tar and potash, e.g., [2,36]. During the 17th to 19th centuries, in areas close to iron production sites, most wood biomass was used for production of charcoal [37,38]. Moreover, one should not forget the people themselves. In large parts of southern and central Sweden, forests which are today used for production of timber and pulp and almost devoid of permanent human inhabitants, were just over a century ago populated by lots of people [39–41], smallholder farmers and crofters, maintaining their livelihood by growing some crops, keeping some livestock, managing small gardens, in addition working at nearby located manors, mills, mines, or ironworks.

This plethora of historical impacts largely ceased during a period from the late 19th to mid-20th century, primarily due to a modernization of agriculture and the introduction of modern production forestry, but also promoted by industrialization and urbanization [21]. This led to a landscape transformation which can be illustrated by the changing area of semi-natural grasslands. These grasslands were managed by grazing or mowing, they were not much influenced by fertilizers, sowing or plowing, and they usually have a long history of management, for centuries or in some cases even millennia [18,42]. Semi-natural grasslands typically harbor a very high local-scale plant species richness, e.g., [43]. In one study area in southern Sweden, it was estimated that over 96% of semi-natural grasslands in 1900 was lost in 2013, most of them transformed to forest [44]. The area used for forest grazing has decreased with over 98% since 1927 [45].

The forest cover in southern Sweden has increased during the last 100 years [46], and the same holds for the standing biomass of wood in these expanding forests [47,48]. The forests we have today are generally much denser, darker and less disturbed than they have been for a very long time, perhaps ever. Increased shading from a dense tree canopy has been identified as one major driver behind recent changes in the boreal forest understory, together with increased nitrogen deposition and climate warming [49]. There is an ongoing decline of several plant species in the boreo-nemoral forest zone. The species in the forest understory that have decreased most during the 20th century are those with a small stature, i.e., low-growing species that are particularly sensitive to increased shade, either from the canopy or from taller understory species [50].

As a consequence of these changes in the boreo-nemoral forest environment, we may expect that there are floristic legacies of the historical, no longer occurring, human use of the forests (Figure 1). The main objective of this paper is to review and synthesize studies on floristic legacies in Swedish boreo-nemoral forest understory.

The rationale for studying floristic legacies is that they contribute to the biological diversity in forest ecosystems, and that they can be regarded as biological cultural heritage, e.g., [17,34,51]. In addition to being interesting in their own right, such legacies are also informative for developing appropriate conservation plans, e.g., [52], and they may be useful for predicting the future of forest ecosystems, e.g., [53,54]. The paper ends with a brief discussion on the question if, and if so, how, these legacies should be preserved.

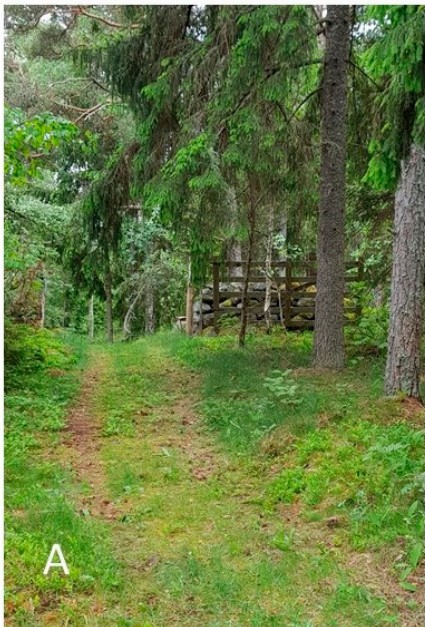
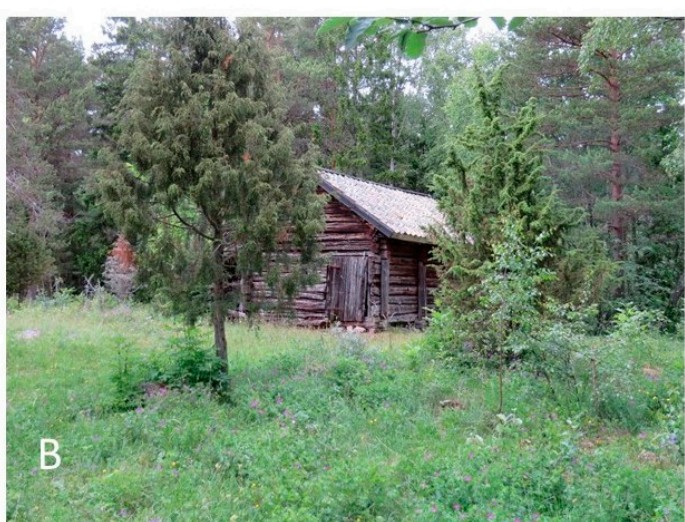

**Figure 1.** The background of floristic legacies in boreo-nemoral forests. (**A**) A gate in a stone wall enclosing a former hay-meadow on Singö, province of Uppland, Sweden. Forest grazing was conducted in the foreground of the gate and the stone wall. The site is slowly reverting back to forest. (**B**) An old storage building for hay in the meadow enclosed by the stone-wall in (**A**). The grass sward still maintains a rich grassland flora. (Photos: the author, July 2022).

This paper also has a second objective. The species comprising the floristic historical legacies in boreo-nemoral forest understory can be seen as belonging to three different categories. Firstly, there are species which have their main occurrence in open grassland systems, and which maintain populations in the current forest environment. Secondly, there are species which are not found in open grassland systems—they are true forest species—but they were nevertheless favored by the historical impacts creating more open, lighter, and disturbed forests. As the first group of species so far has received more attention than the second, this paper also contains a case study comparing two species belonging to this latter category.

A third category of floristic legacies is cultivated plants at forest settlements such as crofts and small farms, abandoned during the period of transformation of forest land use between the late 19th and early 20th century. Many of these cultivated plants are still found at the same sites, sometimes as the only remaining trace of the former settlement. Such cultivated plants, the most common being *Ribes uva-crispa* L. (gooseberry), *Syringa vulgaris* L. (lilac), and *Malus domestica* Borkh. (apple), are often encountered in boreo-nemoral forests, e.g., [41,55]. However, cultivated garden plants are beyond the scope of this paper, and are not treated further.

Before reviewing evidence regarding floristic legacies, and presenting the case study, below follows a brief overview of ecological mechanisms behind floristic legacies, and some remarks on methodological issues.

## 2. Ecological Mechanisms behind Floristic Legacies

In Longman Dictionary of Contemporary English, legacy is defined as "a situation that exists as a result of things that happened at an earlier time". Applied in the context here, this mean that some previous (henceforth "historical") ecological condition—such as any of the plethora of effects mediated by human actions listed above in the Introduction—have promoted establishment or an increasing population of a certain plant species. Furthermore, in order to be meaningful, identifying a species as a legacy presumes that the species would not exhibit its current occurrence, unless the historical ecological condition had existed.

Given this broad definition, there are two main mechanisms behind floristic legacies. Firstly, the species still persists even though the particular ecological conditions necessary for its establishment and population growth have ceased to exist. For example, we may envisage a species that established at a site as a direct result of effects of livestock grazing in the forest, the species would otherwise not have been there, but the species still persists at the site even though livestock grazing has for long ceased.

The second mechanism is when the ecological conditions altered by historical land use still persists, affecting the current flora, even though the agents causing the altered ecological conditions have ceased to exist. For example, we may envisage a historical management regime altering soil conditions and that these soil conditions persist and affect the current plant species composition. Several studies conducted in forest of continental Europe have documented this phenomenon [5,56–58]. Although one could perhaps say the legacy in this case is the ecological condition rather than the species associated it, this semantic issue is here disregarded, and records due to this mechanism are recognized as floristic legacies.

The first mechanism—a species is present despite the ecological conditions favoring it has ceased to exist—is often referred to as "remnant populations" [59,60]. By definition, a positive population growth rate ($\lambda > 1$) implies an increasing population, while $\lambda < 1$ implies that the population is decreasing, and that it will ultimately go extinct. Incorporating a spatial component, maintenance of a local population can be achieved even if $\lambda < 1$, namely if there is a flow of individuals into the population from elsewhere; the local population being a sink population [61]. Many species have a patchy distribution composed of several more or less interconnected local populations, a metapopulation [62]. Some of these local populations may be sink populations, even though the metapopulation as a whole is persistent. However, as legacies refer to a time sequence of events (past and present), one needs to incorporate a temporal component. Eriksson [59] defined the concept remnant population as a local population which is maintained despite $\lambda < 1$, without being a sink population. In order to make sense, this concept necessitates defining a timescale which is considered relevant. Many plants have life stages that can be very long-lived, viewed in a timescale of decades and even centuries. For example, this holds for many clonal plants [63] and for plants with seed banks [64]. Given that this corresponds to a timescale often used when investigating vegetation changes, recognizing remnant populations does make sense.

Remnant populations can be seen as "sink populations in time". While an ordinary sink population is maintained by individuals moving in space from a local population with $\lambda > 1$ to the sink population, remnant populations are maintained by individuals "moving in time" from a temporal stage with $\lambda > 1$ to a later temporal stage with $\lambda < 1$ (Figure 2). Potentially, this may enable the population to again achieve $\lambda > 1$ if the conditions improve, a phenomenon referred to as remnant population dynamics [59]. However, we here focus on the remnant populations as such, having $\lambda < 1$. Species with remnant populations contribute to an extinction debt [65], and the expected time to extinction for a remnant population corresponds to the time it takes for the historical signal to vanish, i.e., the historical time depth for the floristic legacy.

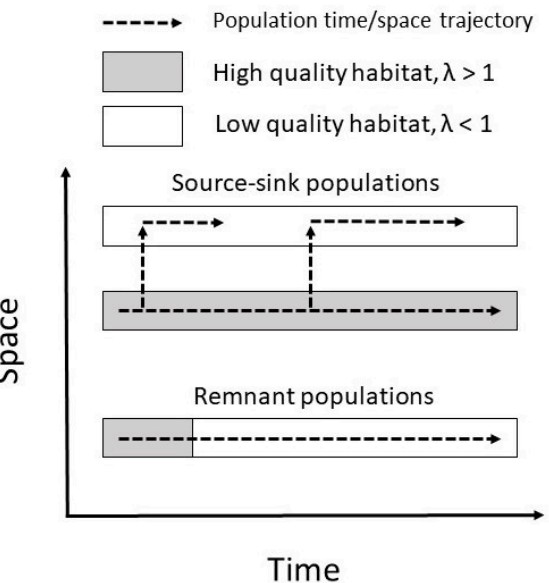

**Figure 2.** A graphic illustration of the difference between source-sink and remnant populations ("source-sink populations in time"). Adapted from Ref. [59].

An additional way of looking at mechanisms behind floristic legacies is to use the niche concept. There are several different definitions of ecological niche (see [66] for an overview). The most common conception is that the ecological niche encapsulates the ecological parameter space within which the species can exist, i.e., have a population growth rate $\lambda \geq 1$ [67]. This is based on Hutchinson [68], who also introduced the useful distinction between fundamental and realized niches. The fundamental niche includes all aspects of the ecological parameter space in the absence of other species, whereas the realized niche is a part of the fundamental niche restricted by interactions with other species [66]. Even though Hutchinson [68] stressed competition as the major interspecific interaction, the factors restricting the fundamental niche may be other interactions, such herbivory, or even abiotic conditions. In addition, a species' fundamental niche may differ among life cycle stages. This has been termed "ontogentic niche shifts" [69]. One example of such ontogenetic niche shifts in boreal forest is understory *Vaccinium* shrubs, where the conditions required for seedling recruitment are partly or fully outside the niche for adults [70]. Indeed, Grubb [71] stressed that understanding the dynamics of plant communities may require that a regeneration niche is recognized, specifying the totality of conditions needed for recruiting new individuals to a population. These conditions may be completely different from the conditions required for an established adult plant to thrive.

By definition, floristic legacies occur because historical land use once made it possible for species to express their fundamental niche at sites where they otherwise would not have been able to establish and survive. The fact that the species still remain there reflects that the current ecological parameter space is within the fundamental niche for at least a part of the plant's life cycle. Understanding the mechanism behind a legacy therefore includes identifying which part of the life cycle that is outside the ecological parameter space, and why this is so.

### 3. Some Remarks on Methodological Issues

Considering that we generally lack time series data on population occurrences covering both the historical conditions and the present ones, there are methodological problems to detect floristic legacies empirically. A record of a plant species in the field (for example as a putative remnant population) may reflect that the species is present because of historical conditions, and has remained at the site long after these conditions have changed, or that the species has dispersed there later, and may be a sink population established from a source population elsewhere.

Distinguishing between these alternatives is difficult. One method is to couple present-day floristic composition with the site's history as revealed by historical maps. This method is commonly used in research conducted in semi-natural grasslands where the impact of historical management on present-day diversity and species composition has been shown in many studies [18, and references therein]. Thus, historical maps, or other sources of historical information, can be used to identify historical ecological conditions in present-day forests, and these can be compared with data on species' current occurrence and performance.

Another approach is to conduct demographic studies combined with modelling, in order to estimate how population growth rate varies with environmental conditions reflecting different land use trajectories, e.g., [72–74]. If no demographic data is available, there are other ways to detect a likely remnant population, for example based on the existence of life-cycle stages enabling population persistence even in the absence of seed production and seedling recruitment, such as subterranean propagating structures and rhizomes, re-sprouting ability, and seed banks [75].

For floristic legacies resulting from persistent ecological conditions (e.g., soil) caused by historical management, the problem is not to associate the ecological conditions with the presence of certain species (this can be studied by experiments). The problem is rather to show that it was the historical management per se that caused the ecological conditions, and that these conditions do not occur irrespective of the historical management.

In light of these methodological difficulties, one needs to accept an element of subjectivity in assessing floristic legacies, based on what is the most reasonable and likely interpretation. This is something the ecologist has in common with historians trying to assess causation in historical processes, e.g., [76]; the conditions or events considered are no longer possible to observe, and experiments can rarely be employed. In the review below, it is thus unavoidable that there is an element of interpretation when assessing whether records of species in forests are legacies of historical land use.

## 4. Grassland Species as Floristic Legacies of Historical Land Use in Forests

Table 1 contains a list of grassland species referred to in the studies reviewed below in this section. In order to avoid over-representing grassland species recorded in forests, a conservative criterion was used. This criterion was derived from a commonly used overview of semi-natural grassland plants in Sweden [77], which lists species according to their dependence of grassland management, grazing or mowing, for maintaining populations, i.e., exhibiting $\lambda \geq 1$. Four categories of grassland species were recognized, defined based on their sensitivity to abandonment of management. The categories A, B and C include species which decline after abandonment, with A being those responding most rapidly, B is intermediate, and C being the least sensitive, responding last. Table 1 only lists grassland plants assigned to the categories A and B. Thus, "grassland plants" henceforth mean species having their core occurrence in semi-natural grasslands.

The strongest evidence for floristic legacies comes from studies of present-day forest sites with a known history as pasture and meadow, and where the studies include control sites with similar forest lacking this history. One study [78] investigated species composition at forest clear-cut sites with a historical background as hay-meadow, based on maps from the late 19th century, and compared them with adjacent sites which were not used as meadow historically. It was found that clear-cuts with a history as meadow had 36% higher species richness. Overall, 48 species were significantly or close to significantly associated with clear-cuts with a history as meadow. Seven of these were species (using the conservative definition) categorized as grassland plants (Table 1). In a similar study [60], species composition was compared between forest sites with a history as meadow (also based on late 19th century maps) and sites which have had a continuous coniferous forest cover. The species composition at the forest sites was also compared with present-day species-rich semi-natural grasslands, managed by livestock grazing. The species-richness was 30% higher at forest sites with a history as meadow. Overall, 77 species were

significantly or close to significantly associated with forests with a history as meadow, and nine of these were species categorized as grassland plants (Table 1). The forest sites with a history as meadow were also more similar to current grasslands than forests with continuous coniferous cover. The authors of both these studies interpreted the results as evidence of remnant populations.

Even if this interpretation seems reasonable, the result that the overall species richness was higher at the clear-cut/forest sites with a history as meadow (i.e., not only for grassland plants) illustrates one of the methodological problems mentioned in the previous section. The sites used as meadow historically may have been chosen by the farmers because they were initially identified as more productive than adjacent land. Alternatively, the meadows were improved as a result of the management, and this also affected species richness. Historical meadows were exceptionally rich in plant species, e.g., [27,79] and management of wooded meadows improved soil conditions because deep tree roots increased the available nutrient pool, nutrients were recycled to the top soil after leaf shedding, and leaf litter created favorable habitat for earthworms [36] (p. 204). Although there is no additional information enabling us to distinguish between these alternative explanations, the association between historical management and current species richness found [60,78] holds as a legacy of historical management.

Combining historical maps with current records of species occurrences can provide a basis for modelling how spatial dynamics of species develop over time. This approach was used to examine the influence of land-use history on the distribution of *Succisa pratensis*, a perennial often found abundantly in semi-natural grasslands with long history of management [72]. The main conclusion was that the current distribution, which included present-day forest where historical management ceased 100–150 years ago, reflected the landscape in the 19th century. Incorporating assumptions of dispersal rates, and assuming "weak" dispersal, led to the conclusion that the current occurrence pattern of *S. pratensis* is strongly affected by the historical (19th century) landscape. Assuming a higher dispersal rate, the historical effect became weaker, but was still present. Overall, the expected time it would take for the regional population of *S. pratensis* to reach something close to an equilibrium with the landscape configuration of habitats was found to be in the magnitude of several centuries. Considering the rate of historical and ongoing landscape changes, *S. pratensis* is thus likely to never reach such an equilibrium.

These results are in line with other studies that have examined to what extent present-day landscape configuration of grasslands is related to site-specific species richness. The prediction from theory is that both site area and connectivity of grasslands have a positive effect of their species richness. Two studies [80,81] conducted in Swedish and Estonian semi-natural grasslands, respectively, found that neither present-day area nor connectivity were related to site-specific species richness. In contrast, when using maps depicting grassland configuration 50–100 years ago, connectivity (both studies) and area [81] turned out to be strongly positively related to species richness. This suggests that there are time-lags in the correspondence between landscape habitat configuration and species occurrences, and that these time-lags are in magnitude of 50–100 years. It was estimated that there was an extinction debt comprising about 40% of the grassland species [81]. Thus, extinction (albeit slow) is currently a dominating process. This was also the conclusion for *Succisa pratensis* [72]. The current dynamics of this species was dominated by population extinction processes.

Demographic analyses of the grassland perennial *Primula veris* was conducted to estimate population growth rate in populations at sites where grassland management had ceased, with varying time since abandonment [74]. A general conclusion was that the time to extinction ranged from decades to centuries. Field studies were employed to validate the results from the demographic analyses, confirming that *P. veris* occurred abundantly even at forest sites where grassland management had ceased in the early 20th century. Thus, as in *Succisa pratensis*, a large fraction of occurrences of *P. veris* in the modern forested landscape most likely consists of remnant populations.

There are also studies with the limitation that they focused on surveys of sites with a known history of human management, but lacking controls and demographic modelling. One study [82] surveyed forest sites with a history as semi-natural grassland, i.e., pasture or meadow, managed until 60–100 ago. The surveys were compared with investigated records from still managed semi-natural grasslands in the same landscape. The objective was to assess which grassland species were still present in the forests, and whether they had declined. Of 67 species for which data allowed analysis, 44 species had declined but were still present at the former grassland sites. Of these, 27 species are categorized as grassland plants (Table 1).

A similar study [32] focused on five selected species (Table 1). For each of these species, the number of patches (a localized occurrence of the species) was counted within a study area of c. 2 km$^2$ comprising both still managed grassland and forest which in 1854 was documented as pasture. For all five species, the majority (between 62% and 91%) of the patches were located in forest. As the study area was quite small, sink populations cannot be ruled out. However, considering that the abundance of patches was much higher in the forest environment, and referring to the time-lags mentioned above [80,81], it is likely that the forest patches were mostly remnant populations.

Another study [83] was conducted on the annual *Rhinanthus minor*, and the two perennials *Campanula rotundifolia* and *Primula veris* using two field study areas, both comprising still managed semi-natural grasslands and present-day forest on land that previously (in the 1940s) were managed as grassland. The species' occurrences were investigated along a hypothetical successional gradient from open localities to localities with full forest cover. The annual *R. minor* rapidly disappeared after abandonment of grassland management, but the two perennials *C. rotundifolia* and *P. veris* maintained populations along the successional gradient, for *C. rotundifolia* even in the present-day closed forest. For both these species the local population size and flowering frequency declined along the successional gradient. This study also included seed addition experiments (estimating germination, but not seedling survival), and the results suggested that both species were capable of germinating also in the closed forest environment. Thus, as with the example above [32], sink populations cannot be ruled out, but the interpretation is that the distribution pattern largely reflects occurrence of remnant populations.

Still another study [84] surveyed land that formerly was used as meadow or field, and after abandonment in the early 20th century now has encroached to forest. This study did not present a complete species list, but listed ten grassland species recorded at the former meadows (Table 1). A notable observation from this study was that two of the grassland species (*Polygala vulgaris* and *Ajuga pyramidalis*) were significantly more abundant at former crop fields than on former meadows. As these crop fields are not likely to have harbored these species at the time when the fields were managed for growing crops, this provides circumstantial evidence that the species have later colonized suitable locations nearby their presumed original occurrence in the meadows. This suggests that some species are able to maintain populations at refuges where they were not present originally, but which are located in the near surroundings of the original sites. This implies spatial dynamics among small suitable sites in the forest environment, promoting maintenance of grassland species, if not exactly at the same site as the historical management, so in the vicinity.

A similar interpretation was made in a study of historical management other than livestock grazing and mowing [85], investigating species richness and species occurrences at charcoal production sites, remaining as so-called charcoal kiln platforms. Charcoal production was massive historically (until early 20th century) in many areas of present-day forest, and charcoal kiln platforms are usually recognizable as circular areas (about 10 m in diameter) with an aberrant flora. It was found that charcoal kiln platforms harbored a higher species richness than adjacent control sites, and several species occurred there significantly more frequent. One group of species which was more frequent at charcoal kiln platforms was low-abundant (occurring at ≤ 9 of the 50 sites investigated) species categorized as grassland plants; 11 species according to the conservative definition (Table 1).

The interpretation was that charcoal kiln platforms, which have different soil conditions than adjacent controls (higher pH and concentrations of P, K, Mg and Ca), and a lower abundance of the otherwise dominating ericaceous shrubs (*Vaccinium* spp., *Calluna vulgaris* (L.) Hull, provide small-scale refuges for grassland species which historically occurred in the semi-open, disturbed, and grazed forests used for charcoal production.

In a companion study [41], some records on remaining grassland plants were made in a re-survey otherwise focusing on cultivated plants at abandoned crofts, housing people formerly working in the region as miners, or as farmers part-time working with for example charcoal production. These crofts were abandoned between the 19th century and early 20th century, and were surveyed for the first time in 1967. The whole survey was repeated in 2019. Four grassland species were recorded both 1967 and 2019 (Table 1), tentatively interpreted as remnant populations.

Thus, in all, Table 1 lists 34 grassland species, selected based on a conservative criterion for having their core abundance in managed semi-natural grasslands, for which the reviewed studies provide evidence that they occur as floristic legacies from historical land use in present-day forests. However, note that for two of the 34 species, the categorization as grassland plants has been questioned (see Table 1). The strength of evidence is variable, both because of the different design of the studies reviewed, and because of the varying number of studies with evidence for each individual species.

As the categorization used for grassland plants [77] was based on the species' presumed more or less rapid disappearance after abandonment, it may seem contradictory that these species maintain occurrences even long after abandonment of grassland management at sites which now has turned into production forests. However, even if the predicted rate of decline of these species after abandonment of grassland management may have been exaggerated, the categorization of these species as having their core occurrence in semi-natural grasslands is valid.

Relaxing the criterion, for example including category C [77] would have increased the number of species listed as putative floristic legacies. For example, using this relaxed criterion on the results from Ref. [78] would have increased the number of species from seven (Table 1) to 15. For Ref. [82], the number of species would increase from 27 to 41. In a survey of 50 sites in a boreo-nemoral forest area, there were in total 127 understory species recorded [85]. If figures in this magnitude are used as a baseline, and acknowledging that presenting exact figures is difficult as the surveys were conducted in different study areas, this review suggests that a considerable fraction (in the magnitude of at least a third) of the understory plant species richness in boreo-nemoral forests may consist of floristic legacies of historical forest land use. In addition, ecological conditions remaining from historical land use (meadows, charcoal production) are associated with a higher local species richness, also including other species than grassland plants.

Even with the caveats associated with presenting exact figures in mind, the general conclusion is that boreo-nemoral forests contain numerous floristic legacies of grassland species, reflecting these forests' historical land use, for hay-production, livestock grazing, and for production of charcoal, and that this legacy makes up a large fraction of the present-day plant species diversity in these forests.

**Table 1.** Grassland plants for which there is evidence that they occur as legacies of historical land use in boreo-nemoral forests (Marked with X). The category "grassland plants" was based on [77]. For species marked with * the categorization has been questioned [82,84,85]. "Reference" is according to the number in the reference list.

| Species | References | | | | | | | | | |
|---|---|---|---|---|---|---|---|---|---|---|
| | **[78]** | **[60]** | **[72]** | **[74]** | **[82]** | **[32]** | **[83]** | **[84]** | **[85]** | **[41]** |
| *Ajuga pyramidalis* L. | | X | | | | X | | X | X | |
| *Alchemilla glaucescens* Wallr. | | | | | X | | | | | |

**Table 1.** *Cont.*

| Species | References | | | | | | | | | |
|---|---|---|---|---|---|---|---|---|---|---|
| | [78] | [60] | [72] | [74] | [82] | [32] | [83] | [84] | [85] | [41] |
| *Antennaria dioica* (L.) Gaertn. | | | | | | X | | | | |
| *Anthoxanthum odoratum* L. | X | | | | X | | | X | X | |
| *Campanula rotundifolia* L. | | X | | | X | | X | X | X | |
| *Carex leporina* L. | | | | | X | | | | X | |
| *Carex pallescens* L. | | | | | X | | | X | | |
| *Carex pilulifera* L. | | | | | | | | X | | |
| *Carex spicata* Huds. | | | | | X | | | | | |
| *Cerastium fontanum* Baumg. | | | | | X | | | | | |
| *Danthonia decumbens* (L.) DC. | | | | | X | | | | X | |
| *Euphrasia stricta* J.P. Wolff ex J.F. Lehm. | X | | | | | | | | | |
| *Festuca ovina* L. | | | | | X | | | | | |
| *Festuca rubra* L. | | | | | X | | | | | |
| *Juncus filiformis* L. | X | | | | | | | | | |
| *Lathyrus linifolius* * (Reichard) Bässler | X | | | | | | | X | | |
| *Leucanthemum vulgare* Lam. | | X | | | X | | | | X | X |
| *Lotus corniculatus* L. | X | | | | X | X | | | X | |
| *Luzula campestris* (L.) DC. | | | | | X | | | | X | |
| *Pilosella officinarum* Vaill. | | X | | | X | | | | | |
| *Pimpinella saxifraga* L. | | X | | | X | | | | | |
| *Plantago lanceolata* L. | | | | | X | | | X | | X |
| *Polygala vulgaris* L. | | | | | X | X | | X | X | |
| *Primula veris* L. | | X | | X | | X | X | | | X |
| *Ranunculus acris* L. | X | X | | | X | | | | | X |
| *Ranunculus auricomus* L. | | | | | X | | | | | |
| *Rhinanthus minor* L. | | | | | X | | | | | |
| *Succisa pratensis* Moench | | | X | | X | | | | | |
| *Trifolium pratense* L. | | X | | | X | | | | X | |
| *Trifolium repens* L. | | | | | X | | | | X | |
| *Veronica chamaedrys* L. | X | X | | | X | | | X | | |
| *Veronica officinalis* * L. | | | | | X | | | X | | |
| *Veronica serpyllifolia* L. | | | | | X | | | | | |
| *Viola canina* L. | | | | | X | | | | | |

## 5. Forest Species as Floristic Legacies of Historical Land Use in Forests

As mentioned in the Introduction, it is well-known that tree composition, structure and diversity of present-day boreo-nemoral forests maintain features reflecting their history, e.g., [15,34,51,86], particularly referring to deciduous tree species [35,87–89]. For subordinate trees and shrubs, historical land use affects the present-day composition of a guild of woody fleshy-fruited plants [90]. Species richness of this guild was higher at sites with a history as open grassland, and five species were positively associated with this land-use history (*Lonicera xylosteum* L., *Prunus padus* L., *Ribes alpinum* L., *Rubus idaeus* L., and

*Viburnum opulus* L.). Additionally, for other taxa than vascular plants, current occurrences have been suggested to reflect forest history, for example for fungi and bryophytes [91], insects [92] and birds [93,94]. Time-lags similar to those described above for grassland species have been documented for various epiphytes on oaks, where their occurrence reflects the historical rather than the current composition and distribution of the oak stands [95,96].

Although not much studied, scattered in the literature there are also putative cases of historical cultural impacts on distribution and abundance of forest understory plants. *Actaea spicata* L., *Lathyrus vernus* (L.) Bernh., *Galium odoratum* (L.) Scop., and *Festuca altissima* All. are examples of plants whose occurrences may reflect forest history [92]. The three first (*A. spicata*, *L. vernus*, *G. odoratum*) are possible legacies from historical wooded meadows [27,79,97]. A study of *L. vernus* along its northern distribution range suggested that current occurrences partly reflect forest history [98]. However, for *F. altissima* it may rather be the lack of strong historical management impact of grazing or mowing that promotes its current occurrence [99]. Other tentative examples are plants suggested to have been favored by livestock grazing, e.g., *Lycopodium* spp., *Pyrola* spp., *Goodyera repens* (L.) R.Br. and *Linnaea borealis* L. [100], *Carex digitata* L., *Luzula pilosa* (L.) Willd., *Maianthemum bifolium* (L.) F.W. Schmidt and *Melampyrum* spp. [34], *Cephalanthera rubra* (L.) Rich. [101], and *Pulsatilla vernalis* (L.) Mill. [102,103]. *Diphasiastrum* [*Lycopodium*] *complanatum* (L.) Holub may have been additionally favored by intentional burning [34,104]. Both the latter species have been recorded as declining in southeastern Sweden, with c. 52% (*Diphasiastrum*) and 46% (*Pulsatilla*) between 1929 and re-surveys conducted from the 1990s onward [105]. *Goodyera repens* and *Pulsatilla vernalis* are red-listed in Sweden as Vulnerable (VU) and Endangered (EN), respectively [106].

Generally, however, studies of understory forest plants as historical land-use legacies are not as common as for the grassland species reviewed in the previous section. To balance this relative paucity of research, below follows a synthesis of published and new information on two forest understory species, whose present-day occurrences in boreo-nemoral forests are likely floristic legacies of historical land use [17,101]: *Chimaphila umbellata* (L.) Barton (Figure 3) and *Moneses uniflora* (L.) A. Gray (Figure 4). Comparing these two species will also illustrate how differences in the plant life cycle influence species response to a changing forest environment, i.e., responses to the abandonment of historical land use and its replacement with modern production forestry.

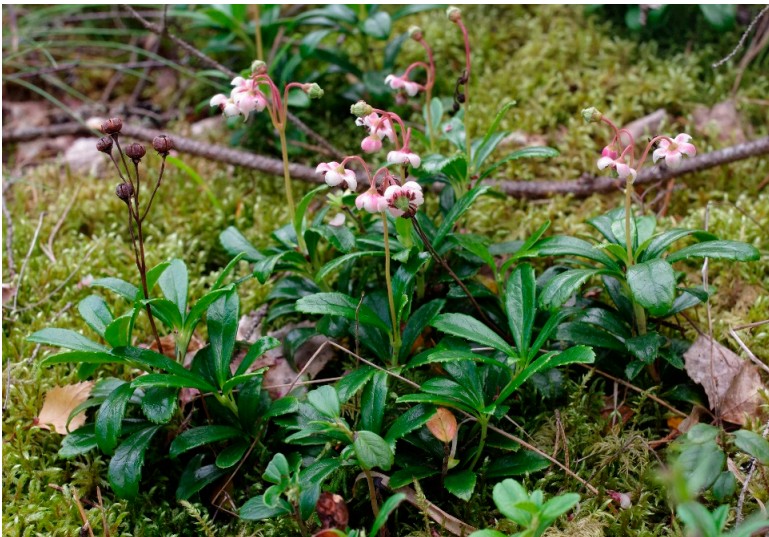

**Figure 3.** *Chimaphila umbellata*, a species favored by historical human use of boreo-nemoral forests, mainly livestock grazing. (Photo: Anna Lundell, July 2013).

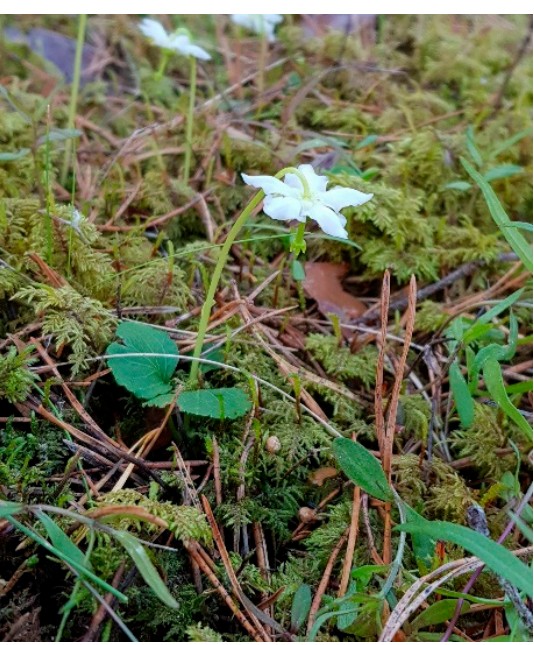

**Figure 4.** *Moneses uniflora*, a species favored by historical human use of boreo-nemoral forests, mainly livestock grazing. (Photo: the author, June 2022).

## 6. The Case Study

### 6.1. Description of the Species

*Chimaphila umbellata* and *Moneses uniflora* (henceforth referred to by their genus name only) belong to the tribe Pyroleae of the family Ericaceae. Within this tribe, *Chimaphila* and *Moneses* are closely related sister taxa [107]. Species of Pyroleae form intricate associations with mycorrhizal fungi with varying degrees of mycoheterotrophy [108], i.e., they derive carbon from a fungal source. As far as known, all species of Pyroleae possess partial mycoheterotrophy (or mixotrophy) [109], except *Pyrola aphylla* Sm. which is fully mycoheterotrophic [110]. This means that they use fungal carbon only during parts of their life, or, if throughout their life, only partially. All Pyroleae have so-called dust seeds [111], i.e., minute seeds produced in vast numbers with a very small and undifferentiated embryo, often lacking any nutrient reserves. Pyroleae have subterranean juveniles which during development, taking place below ground during several years, act as parasites on mycorrhizal fungi [109,112]. Thus, germination and early development from dust seeds of the Pyroleae are completely dependent on fungal associates supplying carbon and other nutrients.

*Chimaphila* (Figure 3) is a perennial evergreen dwarf shrub with leathery leaves, and with an extensive capacity of clonal growth with long creeping subterranean rhizomes [113,114]. It may have over 50 m of interconnected rhizomes belonging to the same clone [115] (clones are henceforth referred to as "genets"). The shoots (henceforth "ramets") are typically 10–20 cm high. Flowering ramets usually have 2–4, sometimes up to 6 flowers. The ramets are iteroparous, i.e., they may produce flowers during several years. It has been suggested that the ramets have a life span of up to six years [113], but considering that they have a woody base and can flower repeatedly, it seems likely that they may reach a higher age. The flowers produce nectar, and they are pollinated by bumblebees [116]. A capsule contains c. 7900 seeds, and a ramet may thus produce over 25,000 seeds [117]. *Chimaphila* utilizes a wide range of fungi during germination, but shows a tendency of becoming more specialized during juvenile ontogeny [118]. Examples of common fungi in developing juveniles were the basidomycetes *Cortinarius* and *Piloderma* [118]. *Chimaphila* is mycoheterotrophic only as juvenile, adults are autotrophic [107,109]. *Chimaphila* has a circumboreal distribution [119]. In Sweden it occurs mainly in the boreo-nemoral zone, most typically in pine forest heaths on sandy soils, but also in mixed pine-spruce forests [114]. *Chimaphila* is red-listed as Endangered (EN) in Sweden [106].

*Moneses* (Figure 4) is a low growing perennial herb, with overwintering green leaves, almost in a rosette, close to the ground. Although the extent has not been studied in detail, *Moneses* is clonal. It has only short rhizomes [120], but spreads clonally by producing ramets from horizontally growing roots [121]. After 2–4 years, probably after reaching over a certain size threshold, a ramet produces one single flower bud, which lives over the winter to produce a flower the following year [122]. After completion of seed production, the ramet dies, without producing any lateral shoots [122]. *Moneses* is buzz-pollinated by bumblebees [116]. A capsule (and thus a ramet) produces c. 7300 seeds [117]. *Moneses* utilizes a range of fungi during germination, and as *Chimaphila,* it shows a tendency of becoming more specialized during ontogeny [118]. Examples of common fungi in developing juveniles were the ascomycete *Meliniomyces* and the basidiomycetes *Russula* and *Tylospora* [118]. Adults have been found autotrophic [109], but other studies suggest that also adults are partially mycoheterotrophic, ultimately becoming specialized on one genus of fungi, *Tylospora* [123]. *Moneses* has a wide distribution in the northern hemisphere, where it occurs in slightly moist coniferous forests with moss-dominated ground cover and in forests with sandy soils, but it is also occasionally found in ditches along road verges and at former fields now covered with Norway spruce (*Picea abies* (L.) H. Karst.) but with only a sparse cover of other field layer plants [105,124]. Thus, it seems able to exploit sites which have been relatively recently disturbed. *Moneses* occurs over almost the whole of Sweden except at high altitudes and in the southernmost region [125].

### 6.2. Population Decline

From the province of Uppland, southeastern Sweden, the occurrences of *Chimaphila* have declined by 81% between 1929 and the 1990s [105]. As being red-listed as Endangered, *Chimaphila* has received some special attention in floristic surveys. A summary was made of the state-of-the-art for Sweden based on surveys conducted from the 1990s onwards [101]. Re-surveys (on average with a 10-year interval) suggested that the majority of the populations were either extinct or had decreased (Figure 5), and more than half of the population extinctions were associated with modern forest management (clear-cuts, plantations) or by increasing dominance of *Vaccinium* and graminoids.

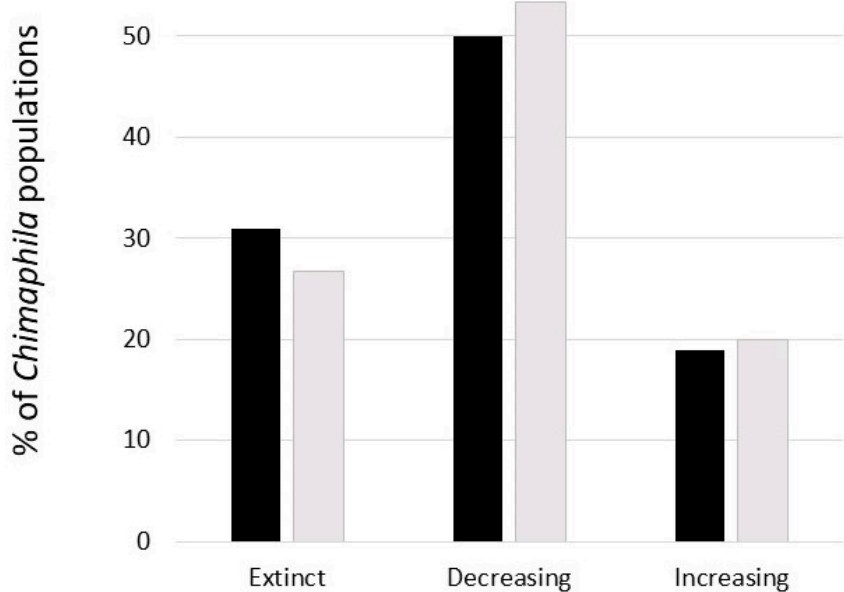

**Figure 5.** The fate of *Chimaphila umbellata* populations based on two sources. Black bars: based on surveys of more than 600 populations from the 1990s onwards and re-surveys on average 10 years later [101]. Grey bars: this study, based on a re-survey in 2021 of 30 populations recorded in 2013 [114].

A study was conducted on a selection of *Chimaphila* populations in two provinces of Sweden, Uppland and Södermanland (southeastern Sweden), based on field studies in 2013 [114]. In 2021, a re-survey was made of these populations (Appendix A, Table A1). The distribution of extinct, decreasing and increasing populations was similar to results for Sweden as a whole (Figure 5), wherefore it is reasonable that the results [114] on the relationships between populations and environmental factors are generally representative for *Chimaphila*. The average number of ramets in 2013 of populations that were extinct 2021 was significantly smaller than those populations still persisting (75 vs. 313; $p = 0.0466$; Appendix A Table A1). Both population size (number of ramets) and fruit-set in *Chimaphila* were negatively affected by competition from dominating *Vaccinium myrtillus* L. and graminoids, and flowering was negatively affected by shading from the canopy [114]. Increasing soil nitrogen was associated with decreasing seed production [114]. Figure 6 shows the forest structure where *Chimaphila* thrives, an open-canopy pine forest with low cover of those otherwise dominating graminoid and *Vaccinium* competitors. This site harbored the largest *Chimaphila* population (>3500 ramets) in the re-survey.

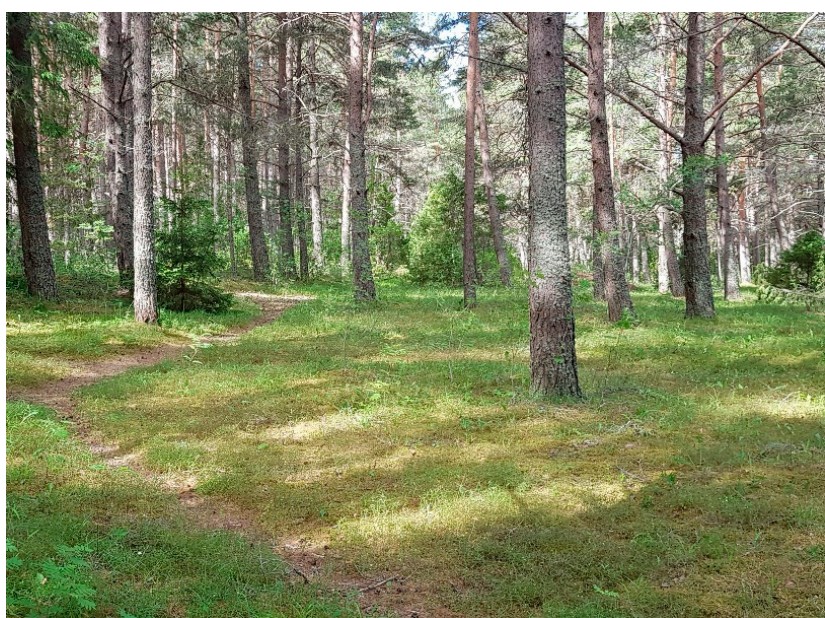

**Figure 6.** A forest where *Chimaphila umbellata* and *Moneses uniflora* thrive. The site at Billudden, province of Uppland, harboring the largest populations of these species recorded in this study. (Photo: the author, June 2021).

The data presented above suggests that *Chimaphila* is currently declining generally, but occasionally have large and even increasing populations. The majority of the occurrences of *Chimaphila* exhibit features suggesting that they are remnant populations, slowly declining from a time where conditions were more favorable.

For *Moneses*, there is less available data. To provide an overview of the development of *Moneses* abundance and distribution, one has to rely on several sources, including a field survey conducted in 2022 in the same two Swedish provinces, Uppland and Södermanland, as was used for *Chimaphila*. There are published floras for both these provinces, useful as a starting point. In Uppland, where the surveys were conducted mainly during 1990s, *Moneses* was recorded as still relatively common [105], although the authors of the flora suggested that *Moneses* is probably declining rapidly. In the flora of Södermanland [124], based on surveys conducted mainly during the 1980s, *Moneses* was recorded as relatively rare. It was remarked that most populations were small, and that *Moneses* has declined sharply ever since the 19th century when it was regarded as common, a decline the authors suggest is due to the abandonment of livestock grazing in forests.

Sweden has a citizen science based information site of species occurrences, "Artportalen" [126], where records of species are reported. The Swedish Association for Botany (Svenska Botaniska Föreningen) in 2015 selected *Moneses* as the "species of the year", encouraging botanists to report finds of the species. *Moneses* may be easy to overlook when it is not flowering and is thus potentially under-reported, but this call is likely to have increased the intensity and accuracy of the reporting.

Finds reported in Artportalen from 2015 onwards, from Uppland and Södermanland, was downloaded. A selection of sites of *Moneses* reported in Artportalen were also re-surveyed in 2022.

There were a total of 133 records of *Moneses* in Artportalen for these two provinces (a record meaning a site-specific finding of *Moneses* at one year during 2015–2021), distributed among 79 different sites. The first author of the flora of Södermanland, Hans Rydberg, one of the most experienced field botanists in this province, on request gave the following information: "I almost never encounter *Moneses* anymore, and when it occasionally happens there are only a few tiny plants" [127]. Furthermore, of all records of *Moneses* downloaded from Artportalen 2015–2021, 75% was either just a notification that this species was present, or a record of a small patch with ≤ 20 flowering ramets.

The results of the re-survey conducted 2022 are summarized Figure 7 and in Appendix A, Table A2. Of the 25 re-surveyed sites, *Moneses* was found at four sites, two interpreted as increasing and two as decreasing. These four were all at sites where *Moneses* was previously (2015–2016) recorded as having a relatively large population (at least 50 flowering ramets, or a total of 450 ramets). All smaller populations were gone in 2022. Among the 21 sites where *Moneses* were extinct, four had been subjected to clear-cutting. Thus, *Moneses* disappears even if the forest still stands. During the surveys, a search was conducted in the surroundings of each of the investigated sites, to look for previously unreported populations. Not one single occurrence was recorded.

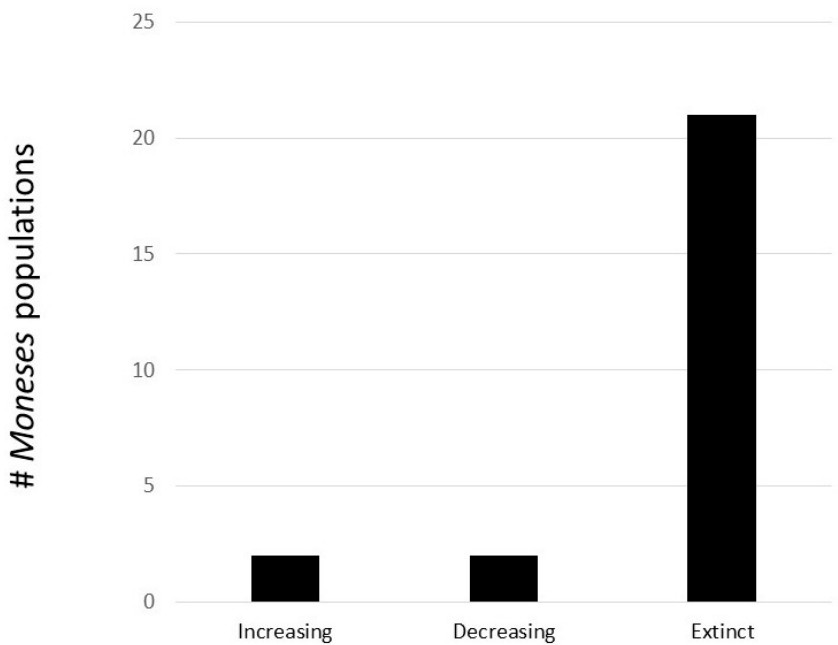

**Figure 7.** The fate of *Moneses uniflora* populations in 2022 in comparison with surveys conducted between 2014 and 2019 (*n* = 25).

The largest *Moneses* population (in terms of flowering ramets) was found at the same location as for *Chimaphila* (Figure 6), a semi-open forests with a limited cover of *Vaccinium* and graminoids. It seems reasonable that large populations of *Moneses* persist because the ecological conditions resemble those which formerly were common due to livestock grazing in forests as remarked in [124].

This data, despite its limitations, suggests that *Moneses* is continually declining. In contrast to *Chimaphila*, where a majority of sites persist several decades even though declining in size, most *Moneses* populations are gone. This difference is accentuated by the fact that the time interval between the surveys for *Moneses* was shorter than for *Chimaphila*. Thus, the capacity of *Moneses* to maintain local populations is seemingly more limited than for *Chimaphila*, and therefore regional persistence of *Moneses* is more dependent on continuous establishment new genets.

In order to understand this difference between the two species, we need to look into their life histories.

### 6.3. Comparison of Life Histories

In order to maintain an established genet, the production of new ramets must balance the loss of ramets dying. A difference between the species is that the ramets in *Chimaphila* live longer. Although no demographic study has been made, the suggestion of a ramet life span up to six years for *Chimaphila* [113] rather seems as an underestimate, as remarked above. In *Moneses*, reproductive ramets die after completion of fruit production. A crude estimate is that this occurs in the ramet's third year of life, on average (B1 in Appendix B), in line with the suggestion [122] that their life span is 2–4 years. In addition, the capacity of clonal propagation in *Chimaphila*, with its extensive rhizome system [115], exceeds the corresponding capacity of *Moneses*, which propagate by horizontal roots. These roots lie near the ground surface [121], which may be the reason why *Moneses* thrives at sites where mosses dominate the ground floor [105,124], i.e., at sites with limited root competition for space. Competing with roots of graminoids, especially, may be difficult for *Moneses*. The ramets of *Chimaphila* also have their leaves placed higher than the rosette of leaves in *Moneses*. Altogether, this implies that genets are larger and considerably more long-lived in *Chimaphila* than in *Moneses*, and that *Moneses* is even more sensitive to competition by potential dominants (*Vaccinium* spp. and graminoids) in the forest understory.

At the genet level, a balanced population requires that each reproductive genet on average produces one new genet during it life. Both *Chimaphila* and *Moneses* produce a vast number of tiny dust seeds, but *Chimaphila* has the highest seed production. The yearly seed production per seed producing ramet is about 3.4 times higher than in *Moneses* [117], and to this one must add that *Chimaphila* ramets can reproduce during several years.

It might be expected that these tiny dust seeds have long dispersal ranges, but in fact over 95% of dust seeds are deposited within 5 m from the seeds source [117]. Thus, the likelihood of a seed finding a suitable site for recruiting a new genet at another location is very small, despite this vast seed production. For both species, suitable sites for germination and juvenile growth depend on presence of fungi, from which the growing juvenile is totally dependent on for extracting resources. Using field experiments at what was judged as potentially suitable sites, seed germination and growth of subterranean juveniles was investigated over three years [118]. *Moneses* had a higher germination rate than *Chimaphila*. Of germinated *Chimaphila* seeds, only 4.2% reached any of the larger size classes of subterranean juveniles. In contrast, 51% of the germinated *Moneses* seeds reached the larger size classes.

One may speculate that the higher growth capacity of the subterranean juveniles in *Moneses* is due to this species ultimately becomes more specialized in its use of fungi: *Tylospora* spp. [123]. Despite its disadvantage in reducing the range of potential hosts, specialization usually implies increasing efficiency of the parasitism [128,129]. For example, the full mycoheterotroph *Monotropa hypopitys* (Ericaceae), specialized on fungi of the genus *Tricholoma* [130], also has a conspicuously high growth capacity of its subterranean juveniles [118].

Both *Chimaphila* and *Moneses* thrive in open canopy forest where the potential understory dominants, *Vaccinium* spp. and graminoids, are not present, or have a limited occurrence (e.g., Figure 6). Historically managed forests created these conditions; wood was harvested without large clear-cuts and livestock grazing counteracted the expansion of

the potential understory dominants [34]. Both species suffer in modern production forests, which are darker and denser. The suggested interpretation is that the present occurrences of both *Chimaphila* and *Moneses* are legacies of historical use of forests, the most important being grazing by livestock.

However, there is a difference in how these legacies are manifested. In comparison with *Chimaphila*, the life span of both ramets and genets in *Moneses* is shorter; its growth form with rosettes close to the ground and clonal spread by horizontal roots make it more sensitive to expansion of competitive dominants in the understory. This makes adult *Moneses* populations less capable of persisting under deteriorating conditions. On the other hand, if seeds are deposited at a suitable site, *Moneses* may be more capable at recruiting new genets than *Chimaphila*, due to the higher germination rate and growth capacity of its subterranean juveniles.

The current occurrence of *Chimaphila* shows all signs of being composed mainly of remnant populations (Figures 2 and 5). While still being present at many locations, most of these populations are in state of slow decline. When one encounters *Chimaphila* in the field, it is likely that the ecological conditions necessary for persistence of this species are no longer existing at that particular site. In contrast, *Moneses* is more sensitive to deteriorating local conditions, and show signs of responding more rapidly. *Moneses* appears more like a "fugitive" species with transient occurrences, exploiting sites (at least in any significant numbers) which still maintain ecological conditions similar to those which historically prevailed at the time when livestock grazed the forests. However, if the niche requirements for adults are expressed at only few sites in modern production forests, the likelihood that *Moneses* manage to disperse to and recruit at these sites is extremely small. As also several of the large populations from the earlier records (2015 onwards) were gone in 2022, as indicated by the re-survey data (Appendix A, Table A2), *Moneses* runs the risk of complete disappearance, and at a faster rate than *Chimaphila*.

## 7. Discussion

The main conclusions from this review are the following: (i) There are numerous floristic legacies of historical land use in the understory of boreo-nemoral forests, and these legacies make up a considerable fraction of the current understory plant species richness. (ii) The species considered are both those which have their core occurrence in open grassland systems and species which are regarded as true forest species, but which were favored by historical land use. (iii) Many of the species that occur as legacies have remnant populations *sensu* [59], but there are also species whose occurrence reflects the existence of ecological conditions resembling the historical forest structure, although the agents causing these ecological conditions are no longer acting. (iv) There are cases where local-scale species richness, taken as a whole, is a legacy of historical activities in forests. (v) Two of the forest species appearing as floristic legacies are *Chimaphila umbellata* and *Moneses uniflora*. Both species are declining, but due to differences in their life histories, it is expected that *Chimaphila* will maintain its populations longer despite deteriorating conditions, while *Moneses* is more dependent on local conditions resembling what was prevalent historically.

In light of the history of Swedish boreo-nemoral forests, briefly summarized in the Introduction, these legacies reflect that forests for not so long time ago (less than a century) were used very differently from today. Smallholder farmers were still relatively common in forest landscapes until the post-World War II rationalization of Swedish agriculture [131]. Charcoal production occurred to some extent until the 1950s [37]. Livestock grazing in forests were still occurring regionally at least until the 1970s [34]. Although forestry with plantations and clear-cutting practices started to become used by large forest owners (state and private) in northern Sweden during the first part of the 19th century [132], selective cutting was not fully replaced with clear-cutting before the mid-1900s [133]. In southern Sweden, where more forest was privately owned by farmers, this shift may not have been fully implemented until the 1970s [134]. Thus, the conditions characteristic of

boreo-nemoral forests historically were quite common until no more than 50–60 years ago. Especially on less productive land, these features take time to vanish, even if the previous management practices have ceased to exist. This is the general explanation for the floristic legacies presented in this review.

For the grassland plants appearing in forests as legacies, the most reasonable explanation is that a part of the former species-pool of the managed grasslands remain at the same location, or in its vicinity, as remnant populations. As forest grazing by livestock covered the largest area of historical forest management regimes [34], this stands out as a dominating factor. The effect of forest grazing was that open-habitat species occurring in the surrounding landscape expanded into forested areas, where they to some extent remain until today [34], i.e., the group of species listed in Table 1. Additionally, for the true forest species, conditions caused by livestock grazing stands out as the dominating explanation behind these species' presence as legacies. Forest grazing has been suggested to increase the local plant species richness in forests [34,135,136] and one contemporary study from Finland found such a relationship [137]. However, as mentioned previously, the effects on the flora of livestock grazing in boreal and boreo-nemoral forests have not been much studied. Considering the plethora of studies of grazing effects in various grasslands [18], this lack of research may seem puzzling.

The reason why many forest species were favored by forest grazing by livestock is that such forests were lighter (semi-open canopy), with small-scale disturbances, and less dominance of *Vaccinium* shrubs and graminoids. Similar effects may derive from intentional burning to improve grazing conditions [25,100]. These conditions are expected to allow small-statured, less competitive species to thrive. One may tentatively suggest a guild of forest understory species whose current occurrences at least regionally may be legacies of forest grazing, for example including *Lycopodium* spp., *Diphasiastrum complanatum*, *Pyrola* spp., *Linnaea borealis*, *Maianthemum bifolium*, *Pulsatilla vernalis*, the orchids *Goodyera repens* and *Cephalanthera rubra*, and the two species in the case study, *Chimaphila umbellata* and *Moneses uniflora*. Further studies are however needed to examine the extent to which these species (and others) should be regarded as legacies of historical livestock grazing in boreo-nemoral forests.

The case study focusing on the two latter species was motivated by the lack of such studies. At least in the provinces examined, Södermanland and Uppland, the results suggest that both species appear as legacies. Both species are in a process of ongoing decline, and for *Chimaphila* the majority of sites investigated harbor remnant populations. For *Moneses*, the interpretation was that declining populations (i.e., where the local ecological parameter space is partly outside this species' fundamental niche) disappear quite rapidly. The more limited clonal ability of *Moneses* and the short life span of ramets do not promote survival of declining adult genets for long under deteriorating conditions. It may seem unlikely that dispersed seeds of *Moneses* are able to find suitable sites for establishing new populations to compensate for genet mortality. Instead, maintenance of local populations of *Moneses* depends on ecological conditions resembling those which were common in historical forests, semi-openness and a sparse cover of *Vaccinium* spp. and graminoids, and a ground dominated by mosses.

It may also be that the mycorrhizal fungi associated with *Chimaphila* and *Moneses* have declined. The diversity of ectomycorrhizal fungi has been found to decrease with increasing nitrogen load [138,139], and several of the mycorrhizal fungi most important to *Chimaphila* and *Moneses* depend on decaying logs, for example *Tylospora* spp. (especially important for *Moneses*), *Piloderma* spp., and *Russula* spp. [140,141]. As the amount of dead wood is reduced in production forests [142], this also indirectly affects *Chimaphila* and *Moneses* negatively. One can speculate that similar mechanisms operate in spore forming plants with non-photosynthetic subterranean and mycorrhizal gametophytes, e.g., *Lycopodium clavatum* [143], which in addition to ectomycorrizal fungi associate with Glomerian AM fungi [144]. AM fungi are typically considered not to be common in boreal forests, e.g., [145,146], but a considerable diversity of AM fungi was discovered in boreo-

nemoral forests with a species-rich understory of herbs [147], probably associated with these forest's former use as wooded meadows. Thus, the historical use may have opened niche space for AM fungi in boreal and boreo-nemoral forests, thus favoring plants such as *L. clavatum* dependent on these fungi.

If the main conclusions of this review are correct (there are lots of floristic legacies in understory of present-day boreo-nemoral forest), what are their implications? Is it important, and if so, for what?

These legacies will ultimately disappear under current forest management regimes, even though it may take a long time. It is easy to argue that floristic legacies of historical land use are interesting in their own right, as biological cultural heritage [17,51], and also just because one of the basic aims of ecology is to understand the mechanisms behind the occurrence and distribution of species. For forest ecosystems, historical aspects have often been neglected, at least in the context of conservation biology [17,19]. If nothing else, this paper's main conclusion implies that the historical use of boreo-nemoral forest is a necessary component of any study aiming to understand and explain the current biodiversity in these forests. A focus only on contemporary ecological conditions will not be sufficient. Much of the focus in forest conservation revolves around preserving untouched forests in nature reserves and National Parks. However, if the history of human use of forests is neglected, this may lead to unexpected decline of many forest species even if the forest are protected from clear-cutting and logging [34].

Many of the legacies are species that have their core occurrence in ecological systems which are not forest, for example open grassland systems. They can be considered as being left behind when management changed during the last 50–100 years. Vellend [148] pointed out that biodiversity research often has an unfortunate value-driven bias, which for forest conservation biology may imply that species that are not seen as belonging to the natural forest ecosystems are overlooked. Obviously, such a value-driven bias would lead to very misleading conclusions on current patterns of forest biodiversity.

For true forest species, such as the ones in the case study, *Chimaphila umbellata* and *Moneses uniflora*, insights on their dependence on historical use of forests comes out as more essential, i.e., if we strive at preserving these species for the future. In contrast to species which have the core abundance elsewhere, these species will not survive unless the ecological conditions of the forests permit them to do so. This is a key question for conservation biology in forest ecosystems. One currently much discussed method is retention forestry, where parts of forest stands are left unlogged [149]. Overall, the focus of retention forestry has not been on forest understory plants whose occurrence reflect historical land use in forest, e.g., [150]. Some studies suggest that the positive effects of retention forestry on maintaining understory richness is quite limited, e.g., [151], whereas other studies suggest that this method is generally positive for maintaining species diversity, also of understory plants, including open-habitat species [152]. For example, stands of the large shrub *Corylus avellana* L. are commonly encountered in forests which were previously used a wooded meadow, and these stands have been found to associate with a high understory species richness [153]. Retaining such stands would thus have positive effects also for open-habitat understory plants.

Other actions would be to identify and preserve forest stands which still maintain features of former forests, i.e., a relatively open canopy of trees with variable ages, and introducing livestock grazing [34]. Unfortunately, this is probably not easily compatible with commercial forestry, at least as long as production forestry focuses on wood biomass rather than wood quality.

A final remark is that if conditions improve (from the perspective of the species), the existence of floristic legacies may function as foci for expansion and colonization of new sites by the species, e.g., [102]. Thus, knowledge of the historical background behind present-day biodiversity patterns may be important for future biodiversity, for example in the context of climate warming, e.g., [53]. Research on floristic legacies of historical land use in boreo-nemoral forests are thus not only providing examples of biological cultural

heritage and, for some species, knowledge useful for conservation, it may also turn out as relevant for understanding the future of biodiversity in these forests.

**Funding:** This research received no external funding.

**Institutional Review Board Statement:** Not applicable.

**Informed Consent Statement:** Not applicable.

**Data Availability Statement:** All data for the review are from the original publications which can be found in the reference list, or presented in Appendixes A and B.

**Acknowledgments:** I am grateful to L. Glav Lundin for field assistance and to A. Lundell for access to survey data.

**Conflicts of Interest:** The author declares no conflict of interest.

## Appendix A

**Table A1.** Results from a re-survey conducted in 2021 of *Chimaphila umbellata* populations recorded in 2013 [114]. In 2013, 38 sites were investigated, among which *Chimaphila* was found at 34 sites. These 34 sites were re-surveyed in July 2021. The exact location and identity of four of these sites were however uncertain and they were therefore excluded. Of the 30 remaining sites, *Chimaphila* was not found at eight sites, and the populations were interpreted as extinct. The survey-data is listed below, divided into those sites where the population had increased (Site 1–6), decreased (Site 7–22), and was extinct (Site 23–30). The average number of ramets in 2013 in still persistent populations (Site 1–22) was 313 (range 7–1776), and in extinct populations (Site 23–30) it was 75 (range 1–314) (Z = 1.99, *p* = 0.0466; Mann–Whitney *U*-test). 'Change' is the ratio #ramets 2021/#ramets 2013.

| Populations Persisting 2021 | | | | | | |
|---|---|---|---|---|---|---|
| **Site** | **Name** | **Coordinates WGS84** | | **#Ramets 2013** | **#Ramets 2021** | **Change** |
| | | **Lat** | **Long** | | | |
| **Increasing populations** | | | | | | |
| 1 | Bollmora | 59.239331 | 18.235026 | 7 | 10 | 1.43 |
| 2 | Munsö Ekerö | 59.414600 | 17.584391 | 339 | 561 | 1.65 |
| 3 | Snöbergen | 59.544097 | 17.968348 | 13 | 15 | 1.15 |
| 4 | Lapp | 60.115844 | 18.659704 | 51 | 133 | 2.61 |
| 5 | Billudden | 60.637188 | 17.461401 | 1208 | 3578 | 2.96 |
| 6 | Notören | 60.567220 | 17.441189 | 437 | 788 | 1.80 |
| Average change | | | | | | 1.93 |
| **Decreasing populations** | | | | | | |
| 7 | Tullgarn | 58.978209 | 17.601575 | 1776 | 592 | 0.33 |
| 8 | Svarvartorp Ekerö | 59.303610 | 17.701306 | 96 | 11 | 0.11 |
| 9 | Södergården | 59.184542 | 18.366396 | 98 | 75 | 0.76 |
| 10 | Lanan | 59.178118 | 18.323817 | 147 | 64 | 0.44 |
| 11 | Lilla Gräskärret 1 | 59.676369 | 17.558602 | 900 | 647 | 0.72 |
| 12 | Lilla Gräskärret 2 | 59.677099 | 17.557752 | 218 | 48 | 0.22 |
| 13 | Bålstaåsen | 59.575964 | 17.521484 | 140 | 133 | 0.95 |
| 14 | Rosersberg | 59.580045 | 17.892885 | 45 | 4 | 0.09 |

**Table A1.** *Cont.*

**Populations Persisting 2021**

| Site | Name | Coordinates WGS84 | | #Ramets 2013 | #Ramets 2021 | Change |
|------|------|------|------|------|------|------|
| | | Lat | Long | | | |
| 15 | Tilskogen | 59.609193 | 17.751514 | 80 | 48 | 0.60 |
| 16 | Österändan | 59.834283 | 18.817657 | 18 | 10 | 0.55 |
| 17 | Långängen | 60.121074 | 18.628193 | 133 | 74 | 0.56 |
| 18 | Hummelfjärden | 60.333915 | 18.422524 | 60 | 21 | 0.35 |
| 19 | Vendelheden | 60.158122 | 17.510899 | 23 | 15 | 0.65 |
| 20 | Billudden/Udden | 60.657148 | 17.508934 | 515 | 286 | 0.56 |
| 21 | Kapplasse | 60.577296 | 17.825108 | 557 | 341 | 0.61 |
| 22 | Slada 1 | 60.532624 | 18.017672 | 20 | 6 | 0.30 |
| Average change | | | | | | 0.48 |

**Populations extinct 2021**

| Site | Name | Coordinates WGS84 | | #Ramets 2013 | | |
|------|------|------|------|------|------|------|
| | | Lat | Long | | | |
| 24 | Åva träsk | 59.162262 | 18.342956 | 5 | | |
| 25 | Ekillabadet | 59.606646 | 17.508787 | 4 | | |
| 26 | Hebbo | 59.515183 | 17.533661 | 314 | | |
| 27 | Jomale telemast | 60.497423 | 18.379444 | 19 | | |
| 28 | Vendelheden 2 | 60.157306 | 17.508011 | 25 | | |
| 29 | Slada 2 | 60.534323 | 18.013214 | 125 | | |
| 30 | Skånsta | 59.491324 | 18.324522 | 106 | | |

**Table A2.** Results from a re-survey conducted in 2022 of *Moneses uniflora* populations.

| Site | Name | Year | Coordinates WGS84 | | Earlier Records | | Re-Survey 2022 | Change |
|------|------|------|------|------|------|------|------|------|
| | | | Lat | Long | #P | #FR/R | #FR/VR | |
| 1 | Idbäcken | 2015 | 60.595086 | 17.591258 | 2 | 700 R | 301 FR + 996 VR | Increasing |
| 2 | Billudden | 2017 | 60.640076 | 17.465936 | 1 | 50 FR | 355 FR + 750 VR | Increasing |
| 3 | Stigenberg | 2015 | 60.590743 | 17.477950 | 3 | 1340 R | 50 FR + 164 VR | Decreasing |
| 4 | Pärlmossen | 2015 | 60.500548 | 17.473303 | 1 | 450 R | 15 FR + 37 VR | Decreasing |
| 5 | Skirkällan | 2018 | 59.311823 | 16.671995 | 5 | 477 R | 0 | Extinct |
| 6 | Saldalen | 2015 | 59.155962 | 16.363755 | 8 | 59 FR | 0 | Extinct * |
| 7 | Strandstuviken 1 | 2016 | 58.719304 | 17.081295 | 1 | >10 FR | 0 | Extinct |
| 8 | Strandstuviken 2 | 2019 | 58.719269 | 17.081241 | 2 | 12 FR | 0 | Extinct |
| 9 | Örstigsnäs | 2016 | 58.727422 | 17.091121 | 1 | 5 R | 0 | Extinct |
| 10 | Dyvikskärret | 2016 | 58.935340 | 17.585522 | 2 | 19 FR | 0 | Extinct |
| 11 | Åtorpsmossen | 2016 | 58.978506 | 17.589190 | 1 | 150 R | 0 | Extinct * |
| 12 | Tullgarnskogen | 2016 | 58.976070 | 17.568387 | 1 | 2 R | 0 | Extinct |
| 13 | Svartbäcken | 2017 | 59.164998 | 18.204130 | 1 | 1 FR | 0 | Extinct |
| 14 | Tistelkullen | 2015 | 59.745384 | 18.335471 | 1 | 2 FR | 0 | Extinct |
| 15 | Kvarntorpet | 2015 | 59.744616 | 18.335733 | 1 | 5 FR | 0 | Extinct |
| 16 | Ladängssjön | 2015 | 59.860466 | 18.528562 | 1 | 20 FR | 0 | Extinct |

**Table A2.** *Cont.*

| Site | Name | Year | Coordinates WGS84 | | Earlier Records | | Re-Survey 2022 | Change |
|------|------|------|------|------|------|------|------|------|
| | | | Lat | Long | #P | #FR/R | #FR/VR | |
| 17 | Stornotsand | 2018 | 60.190373 | 18.778001 | 1 | 15 FR | 0 | Extinct |
| 18 | Finkarbo | 2015 | 60.523351 | 17.544233 | 2 | 50 FR | 0 | Extinct * |
| 19 | Postmästarhage | 2015 | 60.543124 | 17.474053 | 3 | 200 FR | 0 | Extinct * |
| 20 | Mullbro mossar | 2016 | 60.543450 | 17.476038 | 1 | 200 FR | 0 | Extinct |
| 21 | Mararna | 2015 | 60.618052 | 17.598409 | 8 | 908 R | 0 | Extinct |
| 22 | Stadsskogen | 2015 | 59.838504 | 17.622772 | 2 | 2 FR | 0 | Extinct |
| 23 | Byholma | 2014 | 60.115404 | 18.809443 | 1 | 11 FR | 0 | Extinct |
| 24 | Backby | 2019 | 60.178772 | 18.770642 | 1 | 2 FR | 0 | Extinct |
| 25 | Boda | 2017 | 60.194603 | 18.735252 | 1 | 5 FR | 0 | Extinct |

The records extracted from 'Artportalen' (www.artportalen.se, accessed on 5 March 2022) were from 2015 onwards, from the provinces of Södermanland and Uppland. There were many different rapporteurs. The data reported included inconsistencies, since the figure reported for 'size of the population' (number of ramets) may refer either to the number of flowering ramets (which is what most easily can be recorded in the field), or total number of ramets, including also vegetative ramets.

In the table below, sites with several patches have been clustered because it was difficult to repeat the initial survey's delimitation of patches (#P denotes the number of patches in the initial survey). Coordinates refer to a mid-point among patches. #ramets denote either the number of flowering ramets (FR), vegetative ramets (VR), or ramets irrespective of whether vegetative and flowering ramets were distinguished (R). The re-survey recorded the total number of flowering ramets (FR) and the total number of vegetative ramets (VR).

From the province of Södermanland, there were 41 records (in 10 of these, *Moneses* was just 'noted', i.e., no population counts). The records were distributed among 18 sites (some included several patches, and some were visited several years). The number of records for each year was 16 (2015), 9 (2016), 5 (2017), 5 (2018), 3 (2019) and 3 (2020). Nine sites were re-surveyed in June 2022 (Site 5–13).

From the province of Uppland, there were 92 records (in 26 of these, *Moneses* was just 'noted', i.e., no population counts). The records were distributed among 61 sites (some included several patches, and some were visited several years). The number of records for each year was 58 (2015), 9 (2016), 13 (2017), 5 (2018), 4 (2019), 2 (2020), and 1 (2021). A total of 13 sites were re-surveyed in June 2022 (Site 1–4 and 14–22). In addition, three sites known to the author but not recorded in Artportalen were re-surveyed in June 2022 (Site 23–25).

'Change' is the interpretation of the fate of the population. An asterisk indicates that the site was a clear-cut in 2022.

**Appendix B**

B1. The age of flowering in ramets of *Moneses uniflora*.

Data from *Moneses uniflora* populations recorded in 2022 was used to make a crude estimate of the age of flowering for *Moneses* ramets. Note that "age 1" refers to the first year of life, "age 2" to second year of life, etc. The number of flowering and vegetative ramets was counted at sites where *Moneses* occurred in 2022. The fraction of flowering ramets was 0.270 (based on 2668 ramets).

Under the simplified assumption that genets are in a steady state condition, i.e., new ramets exactly replace the ones that flower and then die, and there is no ramet mortality before flowering (after which the ramet dies), the age of flowering becomes $1/0.270 = 3.7$ years (i.e., they flower during their third-fourth year).

This assumption is however not likely to be valid. Genets may be in an expanding phase (relatively more vegetative ramets) or a retarding phase (relatively fewer ramets). Furthermore, it is most likely that some ramets die before reaching the flowering stage. Since no demographic study has been conducted, we can only assume a mortality rate. For example, if 20% of all ramets die before reaching the stage of flowering, the age of flowering becomes 0.8/0.270 = 3.0 years. If 30% of ramets die before reaching the stage of flowering, the age of flowering becomes 0.7/0.270 = 2.6 years.

Acknowledging that the estimate is uncertain, these rough calculations suggest that *Moneses* ramets typically flower during their third year (±one year), i.e., in line the suggestion by Warming [122].

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
