# Peer review of "Floristic Legacies of Historical Land Use in Swedish Boreo-Nemoral Forests: A Review of Evidence and a Case Study on Chimaphila umbellata and Moneses uniflora"

_forests, doi:10.3390/f13101715_

Round 1
Reviewer 1 Report
The paper presents an interesting review on the effects of forest management and land use legacies on plant communities, which are significant but often overlooked in studies on forest biodiversity. The following case studies on Chimaphila umbellata and Moneses uniflora fit well into the context of the paper. However, I suggest shortening it significantly because most parts are rather general descriptions of the species, their survey history, etc., unrelated to forest legacies and land use. Also, there should be more reasoning for selecting these two species for the case studies. Are they indicator species that represent well the state of the whole understory plant community?
If these issues are addressed, I believe this paper would be a good contribution to the Forests journal and useful for forest ecologists.
Specific comments:
60 remove „for“
62 remove “and”
107 I do not agree they were “left behind”. If they managed to stay there for 100 or more years, it means they persisted through the management and ecosystem change
429 remove “for”
Author Response
It is very difficult to reduce the length of the sections on Chimaphila and Moneses, because the arguments build on three parts: (i) that the species are declining, (ii) that the decline is a response to changing forest conditions, and (iii) that current occurrences are legacies. To motivate the field surveys conducted, the background is necessary to understand. In order to understand the difference between species in how these legacies are manifested, one needs to have a clear picture of the life histories of the two species. However, I did remove some details in the survey history, and merged the two sections 6.3 and 6.4, to reduce the length somewhat.
To motivate the choice of these two species, I added two references at the end of Section 5 were the case study is introduced.
I have corrected the linguistic errors on former lines 60, 62 and 429, and replaced “which, so-to-speak, have been ‘left behind’” with “maintain populations”
Reviewer 2 Report
The study is well organized and designed, it is a pleasure to read.
There are some small changes that should be made:
Figures 1, 3, 4, 6 - date of photo taken the year.
Table 1 I would recommend to use the author's name and then the number of references
in the table 1 Rhinanthus minor L. , the L. should not be Italic
According to the world flora database, the official name of Euphrasia stricta is
Euphrasia stricta J.P.Wolff ex J.F.Lehm.
In the introduction part please show the state of the art of the research, why is it timing and needed? As well as clearly the aim, the objectives are clearly described, which is very good.
The references need some attention,
21, 31, 37, 39, 64, 66, 101, 106, 108, 113, 124, 127, 128, 131, 135, 141, 146 there are mistakes ( in some there is a need of making the year bold, in another there are space mistakes)
80 and 92 delete and
Author Response
I added month and year to legends for the photos.
I refrained from adding the authors names in Table 1 since that would disrupt the layout and make it impossible to fit the Table into one page. Besides, this journal uses numbers for references, referring to the reference list.
In Table 1, the authors of Rhinanthus minor and Euphrasia stricta have been corrected.
In the introduction, the state-of-the art is described clearly on lines 35-41, 75-84, and 104-112. Why this study is needed is described on lines 98-103, and 100-112.
The errors in the reference list have been corrected.